# Oligodendrocytes express synaptic proteins that modulate myelin sheath formation

Alexandria N. Hughes [ID] [1] & Bruce Appel[1]

Vesicular release from neurons promotes myelin sheath growth on axons. Oligodendrocytes express proteins that allow dendrites to respond to vesicular release at synapses, suggesting that axon-myelin contacts use similar communication mechanisms as synapses to form myelin sheaths. To test this, we used fusion proteins to track synaptic vesicle localization and membrane fusion in zebrafish during developmental myelination and investigated expression and localization of PSD95, a dendritic post-synaptic protein, within oligodendrocytes. Synaptic vesicles accumulate and exocytose at ensheathment sites with variable patterning and most sheaths localize PSD95 with patterning similar to exocytosis site location. Disruption of candidate PDZ-binding transsynaptic adhesion proteins in oligodendrocytes cause variable effects on sheath length and number. One candidate, Cadm1b, localizes to myelin sheaths where both PDZ binding and extracellular adhesion to axons mediate sheath growth. Our work raises the possibility that axon-glial communication contributes to myelin plasticity, providing new targets for mechanistic unraveling of developmental myelination.

[1] University of Colorado School of Medicine, Anschutz Medical Campus, Aurora, CO 80045, USA. Correspondence and requests for materials should be addressed to B.A. (email: bruce.appel@cuanschutz.edu)

Cellular communication in the central nervous system (CNS) depends on billions of connections. Synapses, the connections between neurons, are plastic structures that grow and change with experience-evoked neuronal activity. In addition, connections form between neurons and oligodendrocytes, the myelinating cell type of the CNS. Myelin sheaths also are plastic structures, mutable in length, number, and thickness[1–3]. Some of this plasticity may be triggered by experience, because motor learning paradigms increase myelination[4,5] and social and sensory deprivation paradigms reduce myelination of relevant brain regions[2,6,7]. What accounts for myelin sheath plasticity? One possibility is that neuronal activity tunes ensheathment, similar to activity-dependent plasticity at synapses[8]. Consistent with this possibility, Demerens and colleagues demonstrated more than 20 years ago that inhibiting action potential propagation with tetrodotoxin (TTX) reduced myelination of cultured neurons[9]. Activity-mediated communication from axons to myelin sheaths could ensure that active neurons receive enough myelin to faithfully propagate impulses toward termini, facilitating circuit function and higher-order cognitive processes[10]. However, almost nothing is known about communication between axons and myelin sheaths.

Neuronal vesicular release promotes sheath growth and stability[11–13]. Upon cleavage of neuronal vesicular release machinery by clostridial neurotoxins, oligodendrocytes formed shorter sheaths on impaired axons and selected unimpaired axons when presented with a choice[13]. Importantly, timelapse imaging revealed that oligodendrocytes retracted nascent sheaths from impaired axons more quickly than from controls[11]. These studies indicate that oligodendrocyte processes can detect and respond to axonal vesicular secretion. How might oligodendrocytes detect vesicular release? Oligodendrocytes express numerous genes encoding neurotransmitter receptors, synaptic scaffolds, transsynaptic adhesion molecules, and Rho-GTPases[14–16] that, when expressed by neurons, endow dendrites with the ability to detect and respond to release at synapses. Similar to their function in dendrites, these proteins may allow oligodendrocytes to sense, adhere, and respond to axonal secretion[17]. Do oligodendrocytes utilize these proteins similarly to neurons to stabilize nascent myelin sheaths on axons?

Determining how neurons communicate to oligodendrocytes to shape developmental myelination requires an experimental model where both axons and oligodendrocytes can be monitored and manipulated during normal myelination. Here, we used zebrafish to examine features of synapse formation between axons and their myelinating oligodendrocytes, including the accumulation of presynaptic machinery, exocytosis at the junction, and postsynaptic assembly. These synaptic features manifest at the axon-glial junction from the onset of myelination at 3 days post-fertilization (dpf) through 5 dpf, when sheaths are stabilized. Surprisingly, we uncovered unforeseen diversity in the synaptic features present at these contacts: exocytosis sites vary in shape and position under sheaths, and the major postsynaptic scaffold PSD95 also localizes with variable position within some, but not all sheaths. To test whether the synaptic features of the axon-myelin interface are important for normal myelination, we manipulated synaptogenic adhesion proteins in oligodendrocytes. When these same proteins are disrupted in neurons, synaptogenesis falters and synapses are abnormal in size and number. Taking a similar approach, we found that oligodendrocytes expressing dominant-negative adhesion proteins form myelin sheaths with abnormal length and number. We focused on one candidate, Cadm1b (SynCAM1), and found that it localizes to myelin sheaths, where its PDZ binding motif is required for myelin sheath growth. Furthermore, by manipulating the extracellular domain of Cadm1b we found that transsynaptic signaling

to axons promotes myelin sheath length. Our work suggests shared mechanisms of synaptic and myelin plasticity and raises the possibility that synaptogenic factors have previously unrecognized roles in developmental myelination.

## Results

**Axons accumulate synaptic vesicle machinery under myelin sheaths.** Vesicular secretion from axons promotes myelin sheath growth and stability[11–13]. However, we know nothing about the temporal or spatial qualities of vesicular communication to oligodendrocytes. To determine where vesicular release occurs to support sheath growth, we investigated the localization of vesicular release machinery in axons that are myelinated in an activity-regulated manner[11]. By imaging live larvae expressing Syp-eGFP or Vamp2-eGFP we tracked the synaptic vesicle proteins Synaptophysin (Syp) and Synaptobrevin (Vamp2) in individual *phox2b+* reticulospinal axons (Fig. 1a, b, b′). At early larval stages, *phox2b+* axons are sparsely covered by myelin sheaths (~15% of length)[11], permitting us to visualize synaptic vesicle machinery in both ensheathed and bare stretches of axons. Syp-eGFP is distributed in axons in a punctate pattern (Fig. 1b), whereas Vamp2-eGFP diffusely labels membrane (Fig. 1b′), similar to previous observations[18]. We quantified Syp-eGFP puncta over the course of developmental myelination in both bare and ensheathed regions (Fig. 1c). By 4 days post-fertilization (dpf), and continuing through 5 dpf, axons accumulate more Syp-eGFP puncta at ensheathment sites relative to bare regions of axon (Fig. 1d). These data indicate that axons cluster synaptic release machinery at ensheathment sites over time, consistent with the possibility that vesicular secretion mediates axon-oligodendrocyte communication.

To test whether synaptic vesicle machinery is functional at ensheathment sites, we expressed an exocytosis reporter, Syp-pHluorin (SypHy)[19], in reticulospinal axons. SypHy is pH-sensitive and quenched within vesicles but fluoresces upon exocytosis into neutral extracellular space. When extracellular SypHy is endocytosed and reacidified, SypHy is again quenched. By acutely blocking reacidification with the v-ATPase inhibitor bafilomycin A1, endocytosed SypHy remains fluorescent and axonal exocytosis "hotspots" accumulate signal[20] (Fig. 2a). We raised embryos microinjected with *Tol2.neuroD:sypHy* DNA and mRNA encoding Tol2 transposase at the 1-cell stage until 4 dpf, the timepoint at which we first detected enhanced Syp-eGFP clustering under sheaths (Fig. 1d). At 4 dpf, we treated larvae with 1 μM bafilomycin or DMSO vehicle and allowed them to behave freely for 1 h before mounting for imaging. SypHy fluorescence is minimal when larvae are exposed to DMSO vehicle, reflecting the net balance of exo- and endocytosis along axons (Fig. 2a). By contrast, bafilomycin treatment reveals numerous exocytic hotspots along reticulospinal neurons, reflecting net exocytic activity over the hour (Fig. 2a). We confirmed that bafilomycin-induced hotspots resulted from exocytosis by performing the same experiment while blocking vesicular release by cleaving vesicular release machinery with botulinum toxin light chain (BoNT/B)[21] or by suppressing exocytosis with dominant-negative Vamp2 (dnVamp2), a truncation that lacks the transmembrane domain[22–24] (Fig. 2b). Both suppress bafilomycin-induced fluorescent hotspots (Fig. 2b, right), indicating that the hotspots reflect exocytosis sites.

To determine if myelin sheaths are sites of exocytosis hotspots, we performed microinjections into *Tg(sox10:mRFP)* embryos at the 1-cell stage and imaged bafilomycin-treated larvae at 4 dpf. SypHy hotspots are abundant underneath myelin sheaths (Fig. 2c) but variable in morphology. Frequently, SypHy hotspots are punctate, resembling the size and shape of the Syp-eGFP fusion

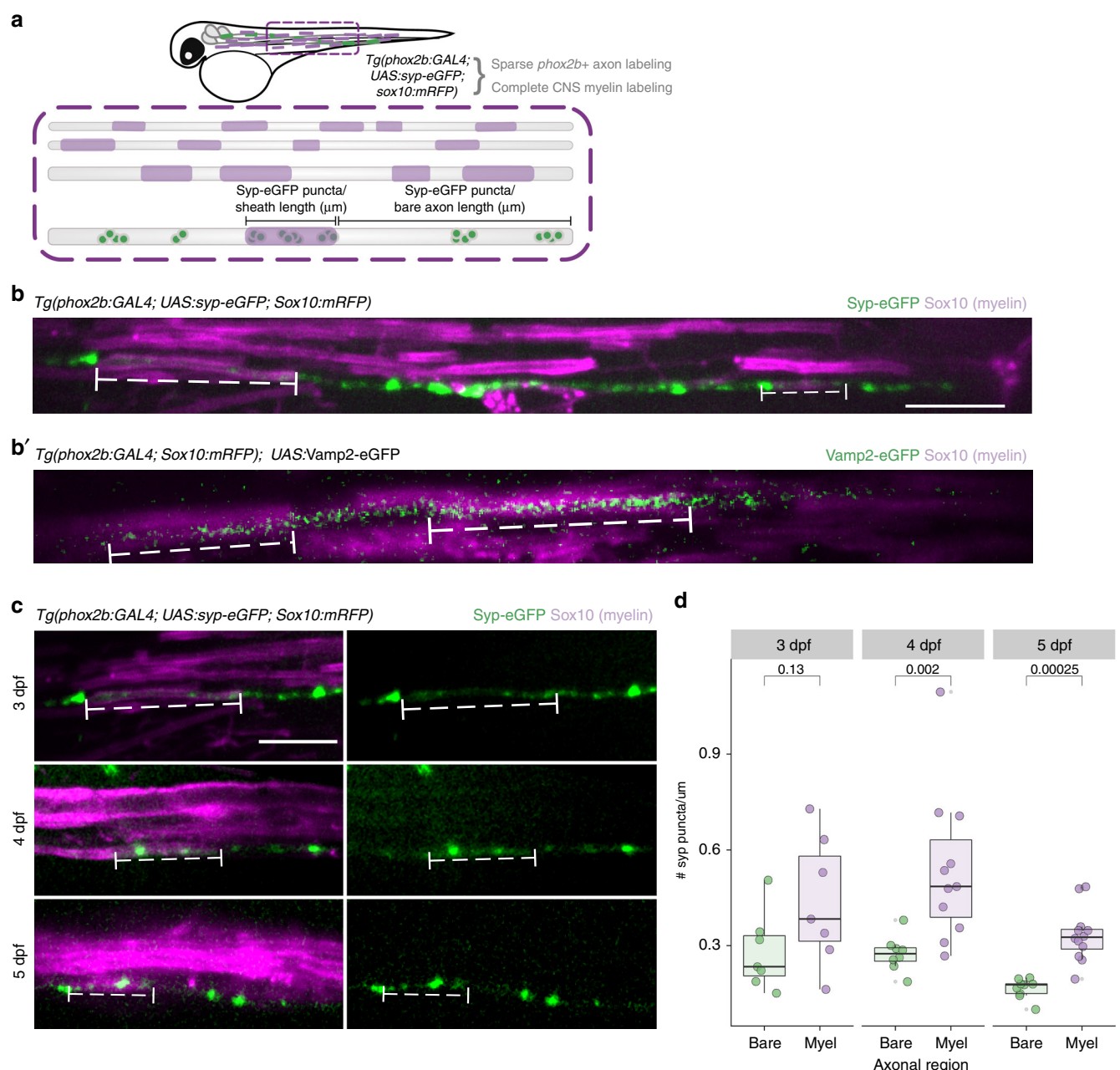

**Fig. 1** Axons accumulate synaptic vesicle release machinery under myelin sheaths. **a** Schematic of transgenes used to assess synaptic puncta density along myelinated and bare regions of axons. **b** Representative image of Syp-eGFP in a myelinated *phox2b* + axon in a living 3 dpf *Tg(phox2b:GAL4; UAS:syp-eGFP; sox10:mRFP)* larva. Bracketed dashed lines indicate two myelin sheaths (rectangles) on the axon. **b′** Same as (**b**) for Vamp2-eGFP; animal genotype is *Tg (phox2b:GAL4; sox10:mRFP)* with transient expression of *UAS:Vamp2-eGFP*. **c** Representative images of Syp-eGFP distribution in myelinated phox2b + axons at 3, 4, and 5 dpf. **d** Comparisons of Syp-eGFP puncta density within bare vs myelinated regions of axons at each timepoint. *n* = 7 bare regions/98 puncta and 7 sheaths/18 puncta (3 dpf); 8 bare/119 puncta and 11 sheaths/43 puncta (4 dpf); 9 bare/75 puncta, and 12 sheaths/38 puncta (5 dpf). Wilcox rank-sum test, *p* = 0.13, *p* = 0.002, *p* = 0.00025 for 3, 4, and 5 dpf, respectively. Scale bars, 10 μm

protein (Fig. 1b) and puncta are almost exclusively located near perinodal ends of sheaths. In addition, some hotspots are restricted to the ensheathed region but diffusely bright underneath the sheath ("filled"), and occasionally axons are entirely illuminated with no visually discernable differences under myelin sheaths ("uniform") (Fig. 2c, d, d′). These categories may reflect different axonal release mechanisms, including synaptic and volume transmission. However, no category is associated with longer sheaths than other categories (Fig. 2e), suggesting that the diversity of release site shape or position does not account for differences in

sheath length. These data are consistent with the possibility that axon vesicle fusion promotes sheath length.

**The postsynaptic scaffold protein PSD95 localizes to myelin sheaths**. We investigated oligodendrocyte expression and localization of the membrane-associated guanylate kinase (MAGUK) postsynaptic density protein 95 (PSD95), a scaffold known for anchoring neurotransmitter receptors at excitatory postsynaptic terminals, to learn if myelin sheaths have

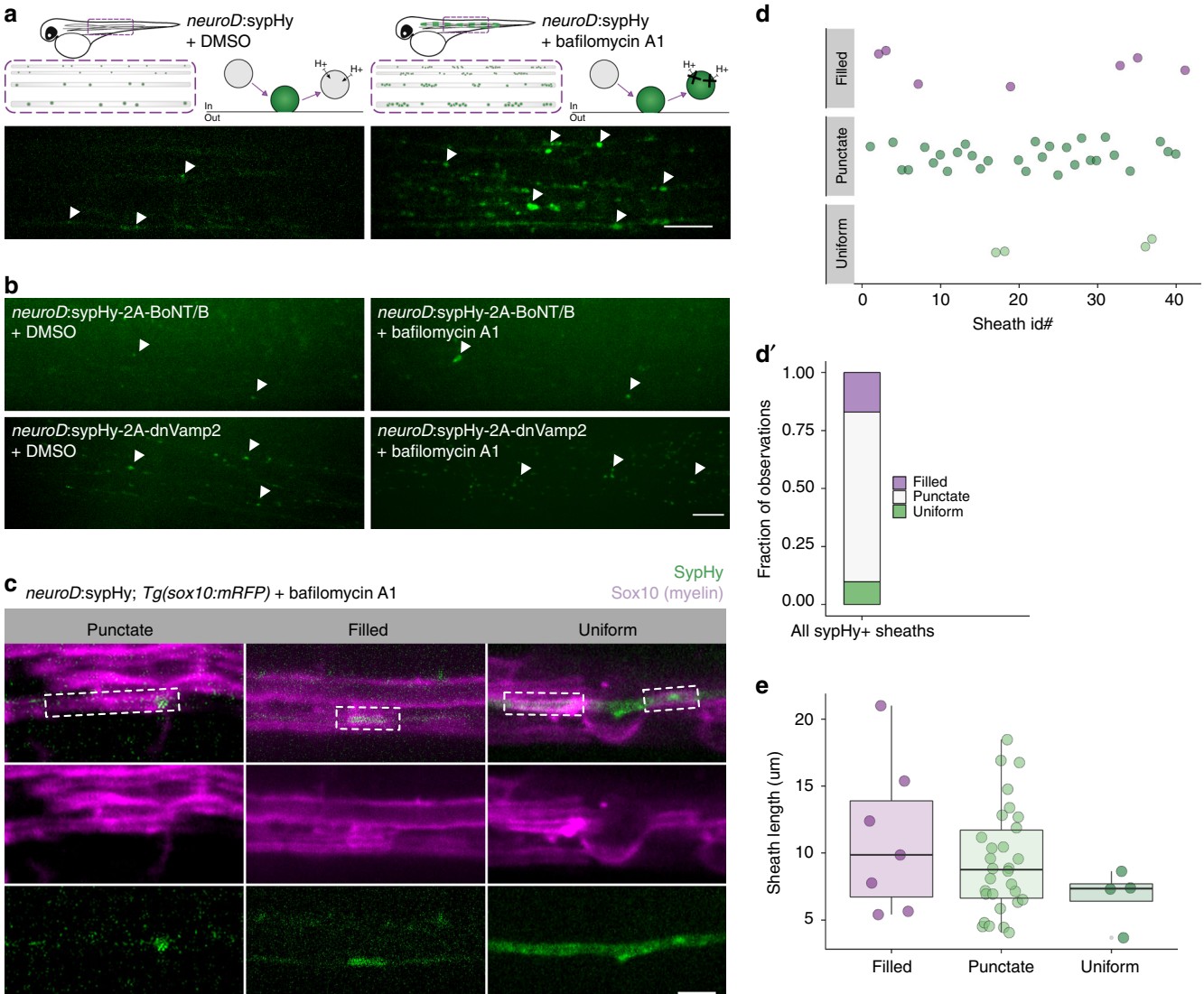

**Fig. 2** Variable synaptic vesicle exocytosis sites under myelin sheaths. **a** Left, schematic of Syp-pHluorin (SypHy) function and transient expression in spinal cord neurons and representative max-projection image of a live 4 dpf larva expressing *neuroD*:sypHy. Arrowheads indicate dim spots of SypHy signal. Right, similar but in the presence of bafilomycin A1 to inhibit vesicle reacidification. Note appearance of several large SypHy + "hotspots" along axons. Scale bar, 10 μm. **b** Similar to **a** except larvae are co-expressing inhibitors of exocytosis, botulinum toxin (BoNT/B) or dominant-negative Vamp2 (dnVamp2). Both BoNT/B and dnVamp2 suppress bafilomycin induction of SypHy hotspots. Arrowheads indicate a few visible SypHy spots. Scale bar, 10 μm. **c** Expression of *neuroD*:sypHy in *Tg(sox10:mRFP)* larvae treated with bafilomycin reveals three types of SypHy-reported exocytosis sites under myelin sheaths: punctate, filled, and uniform. Boxes outline individual myelin sheaths. Scale bar, 5 μm. **d** Classification plot of SypHy signal under 41 sheaths (from *n* = 20 larvae) by category. Presented as a bar in **d'** for visible contribution of each category to the total. **e** No association between category and sheath length, Kruskal–Wallis test

characteristics of postsynaptic terminals. To detect endogenous PSD95 expression, we used a CRISPR/Cas9-mediated GAL4 enhancer trap strategy[25] to generate knock-in transgenic larvae that reports cells expressing *dlg4b*, the gene encoding PSD95. Nearly all *mbpa*:tagRFP + oligodendrocytes also express *dlg4b*, indicating that myelinating oligodendrocytes of the spinal cord express PSD95 (Fig. 3a). As expected, many *dlg4b* + cells do not express *mbpa*:tagRFP, likely reflecting expression of PSD95 in neurons.

We took multiple approaches to label PSD95 to determine its subcellular localization (Fig. 3b, b', b"). First, expressing a PSD95-GFP fusion protein that has been previously used to track PSD95 in zebrafish[26] reveals punctate enrichment at the terminal ends of myelin sheaths and a few sites of enrichment within the lengths of sheaths (Fig. 3b). We additionally labeled endogenous PSD95 by

expressing a genetically-encoded intrabody, PSD95.FingR-GFP, that binds zebrafish PSD95[27,28]. The intrabody contains a transcriptional regulation system that allows unbound FingR to repress further transcription by binding a zinc finger binding site upstream of the regulatory DNA. We first expressed the FingR in oligodendrocytes directly with *myrf* regulatory DNA but without a zinc finger binding site ("unregulated"). The unregulated PSD95.FingR-GFP broadly labels the cytoplasm of oligodendrocytes, with some puncta evident at the ends of sheaths (Fig. 3b'). We then implemented the transcriptionally-regulated system by providing a zf binding site upstream of *UAS* sequence[28] that drives PSD95.FingR expression via co-expression of *myrf*:GAL4 (Fig. 3b"). In contrast to the unregulated system, transcriptional regulation of the intrabody unveils several PSD95 puncta within most, but not all sheaths (Fig. 3b").

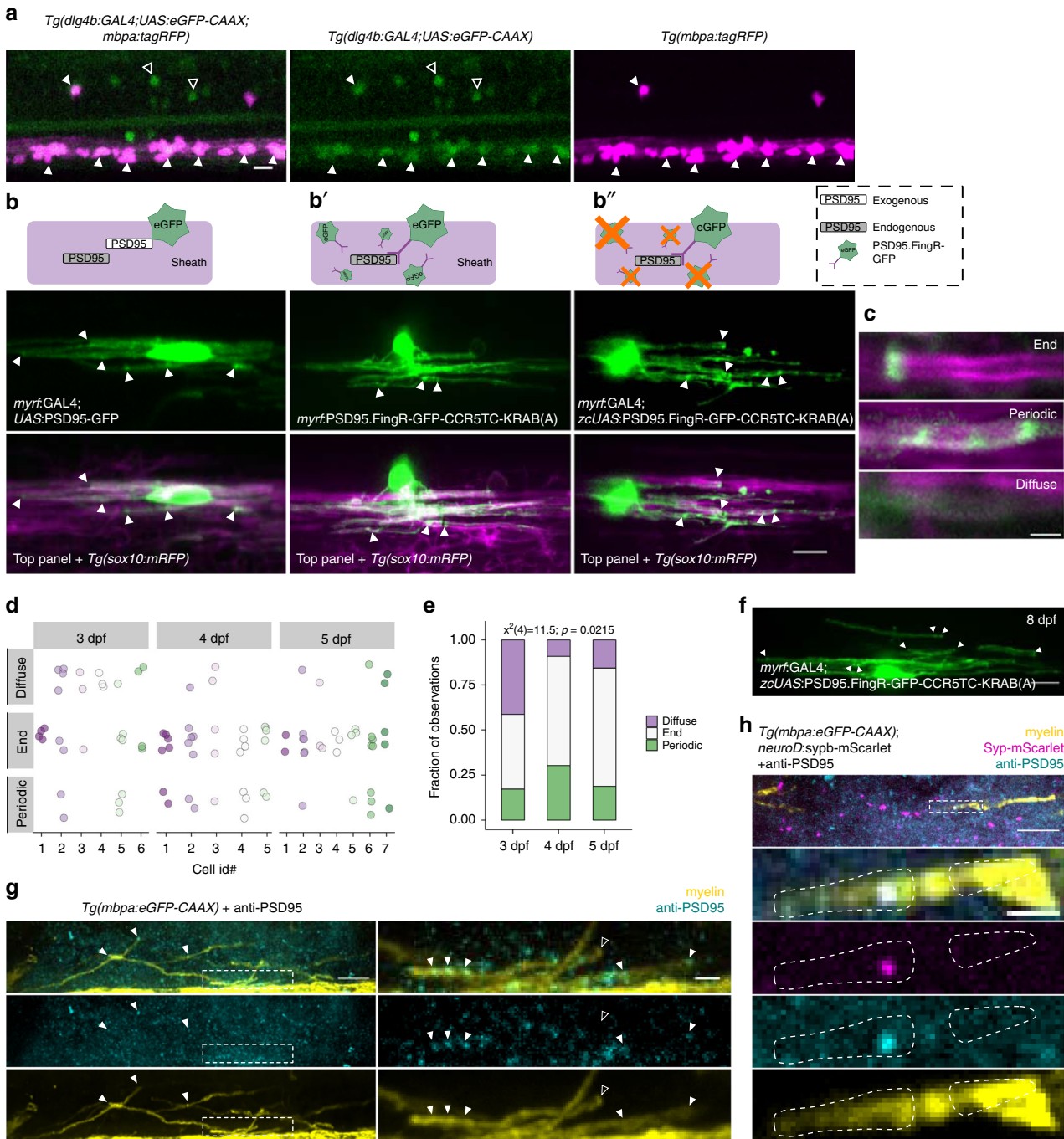

**Fig. 3** PSD95 is expressed by myelinating oligodendrocytes and is variably localized within myelin sheaths. **a** CRISPR/Cas9-mediated GAL4 enhancer trap of *dlg4b* reports spinal cord cells that express *dlg4b* via *UAS*:eGFP-CAAX expression. Larvae additionally carrying the *mbpa:tagRFP* transgene reveal *mbp+* myelinating oligodendrocytes. Closed arrowheads indicate *mbp+*, *dlg4b+* oligodendrocytes and open arrowheads indicate *mbp-*, *dlg4b+* cells. Scale bar, 10 μm. **b**, **b′**, **b″**) PSD95 localization within oligodendrocytes detected by expression of exogenous PSD95-GFP fusion protein (**b**), expression of PSD95. FingR to detect endogenous PSD95 localization (**b′**), and expression of PSD95.FingR to detect endogenous PSD95 localization with a transcriptional repression system to limit unbound PSD95.FingR-GFP (**b″**). Arrowheads indicate puncta, scale bar 10 μm. **c** Expression of the transcriptionally-regulated system (**b″**) with *Tg(sox10:mRFP)* to label myelin shows that not all sheaths display localized PSD95 puncta ("diffuse"), and among those that do, labeling is either restricted to the sheath end ("end"), or periodic along the length of the sheath ("periodic"). Scale bar, 2 μm. **d** Classification plot of sheath labeling patterns (y-axis) for individual cells (x-axis) at 3, 4, and 5 dpf. n = 6 cells/29 sheaths (3 dpf); 5 cells/33 sheaths (4 dpf); 7 cells/32 sheaths (5 dpf). For each age group, dots of the same color denote sheaths belonging to the same cell. Note that most cells formed sheaths with different labeling patterns. Over developmental time, the distribution of labeling patterns changes as assessed by Chi square test (**e**), notably with a reduction of diffuse labeled sheaths, but all three categories persist at 5 dpf. (**f**) Later in development (8 dpf), PSD95.FingR-GFP is less punctate overall but remains enriched at the ends of sheaths (arrowheads). **g** Sagittal section of a 5 dpf *Tg(mbpa:eGFP-CAAX)* larva spinal cord fixed and immunolabeled with anti-PSD95. Right panels show higher magnification of the boxed region. Arrowheads indicate PSD95 signal colocalized with sheaths; open arrowhead marks a sheath without PSD95 at the terminal end. **h** Similar to **g** but with transient expression of *neuroD*:sypb-mScarlet to sparsely label synaptic vesicle puncta. Inset shows colocalized PSD95 and Syp-mScarlet within a sheath. Scale bars are 10 μm and 2 μm (insets)

We next assessed the intrabody localization within sheaths using the regulated form of PSD95.FingR in a *Tg(sox10:mRFP)* line at 3, 4, and 5 dpf. We identified three unique patterns of sheath labeling: puncta restricted to ends of sheaths ("end"), periodic puncta along the length of sheaths ("periodic"), and faint, diffuse labeling with no obvious localization ("diffuse") (Fig. 3c). Most oligodendrocytes have sheaths with variable labeling: some sheaths contain "periodic" puncta, while other sheaths made by the same cell exhibit only "end" puncta. Our raw observations of puncta locations are plotted in Fig. 3d. The most frequent labeling pattern is "end" puncta at every age examined. However, the distribution of localization patterns changes during developmental myelination (Fig. 3e), notably via a reduction in diffusely-labeled sheaths and an increase in end-enriched sheaths. At a later stage in development (8 dpf), there are fewer puncta overall but with continued enrichment at sheath ends and along sheath membrane (Fig. 3f).

To further validate these findings, we labeled fixed tissue sections using an antibody that detects synaptic PSD95 in zebrafish[29]. At 5 dpf PSD95 localizes within myelin sheaths similarly to the end, periodic and diffuse patterns detected by the intrabody (Fig. 3g). To assess anti-PSD95 labeling colocalization with myelin, we performed Costes randomization of the anti-PSD95 signal[30] and found that colocalization does not occur by chance (Supplementary Fig. 1). Thus, three independent methods reveal patterns of PSD95 localization in myelin sheaths, consistent with the possibility that nascent myelin sheaths assemble components of postsynaptic complexes.

Do myelin sheath PSD95 puncta align with axonal synaptic vesicles? To answer this question we sparsely labeled synaptic vesicles with Syp-mScarlet, sectioned larvae in sagittal plane at 5 dpf and labeled sections with anti-PSD95 antibody. This reveals axonal synaptic vesicle puncta in close proximity to myelin sheath PSD95 puncta (Fig. 3h). We note, however, that these coincident labeling patterns are not obligatory because not all myelin sheaths localize PSD95 near axon vesicles. Nevertheless, these data indicate that a subset of oligodendrocyte sheaths accumulate PSD95 close to neuronal synaptic vesicles, raising the possibility that sites of coincident localization are sites of axon-myelin sheath communication.

**Synaptogenic adhesion molecules tune sheath length and number.** In neurons, PSD95 anchors receptors, ion channels, and synaptic signaling molecules at the postsynaptic membrane via its PDZ domains. These domains are found in PSD95 and other synaptic scaffolding proteins, where they bind short, C-terminal PDZ binding motifs located on target proteins destined for synapse localization. PDZ binding motifs direct the localization and function of many synaptogenic, transsynaptic adhesion proteins. This family includes members of the neuroligins, synCAMs, netrin-G ligands, and the leucine rich repeat transmembrane proteins[31]. In addition to adhering pre- and postsynaptic terminals, these proteins are potent signaling molecules. The signaling they induce across the nascent cleft is sufficient to induce synaptogenesis even when ectopically expressed in non-neuronal HEK293 cells[32–35]. Intriguingly, mRNAs encoding many of these synaptogenic adhesion molecules are expressed by oligodendrocytes at levels comparable to neurons[14]. If these molecules are sufficient to confer axonal synapses onto HEK293 cells, could a similar mechanism operate at the axon-oligodendrocyte interface to promote sheath growth and stability?

We identified genes that express synaptogenic proteins in oligodendrocytes by querying published transcriptome databases[14–16] and in-house RNA-seq data[36]. We selected six candidates based on oligodendrocyte expression (Fig. 4a, b) and

the existence of published dominant negatives that disrupt synapse formation when expressed in neurons: Cadm1b (SynCAM1), Nlgn1 and Nlgn2b (Neuroligin1 and −2), Lrrc4ba (NGL-3), and Lrrtm1 and Lrrtm2 (Leucine rich repeat transmembrane proteins 1 and −2). We generated dominant-negative alleles of zebrafish proteins predicted to disrupt PDZ binding[34,37,38] by omitting conserved C-terminal PDZ binding motifs.

We expressed each of these dominant-negative constructs with *myrf* regulatory DNA to disrupt transsynaptic adhesion specifically in oligodendrocytes. To label sheaths, we bicistronically expressed membrane-tethered GCaMP6s-CAAX. We imaged living larvae expressing each of the constructs, or a control *myrf*:GCaMP6s-CAAX (wt) construct (Fig. 4c, d), at 4 dpf to measure sheath number and length. The total sheath length generated per cell, a general measure of myelinating capacity, does not differ between any of the groups (Fig. 4e). However, oligodendrocytes expressing dnCadm1b, dnLrrtm1, and dnLrrtm2 form significantly shorter sheaths, whereas oligodendrocytes expressing dnNlgn2b make longer sheaths (Fig. 4f). Intriguingly, dnCadm1b and dnLrrtm1 oligodendrocytes also elaborate several more sheaths than wildtype (wt), whereas dnNlgn2b oligodendrocytes generate slightly fewer sheaths (Fig. 4g). Because total sheath length per cell is unchanged, this suggests that these adhesion proteins do not influence the ability of oligodendrocytes to generate myelin but rather tune how oligodendrocytes allocate myelin among sheaths. Importantly, for all three dominant-negative constructs that reduce sheath length (dnCadm1b, dnLrrtm1, and dnLrrtm2), expression of the full-length (wt) protein does not reduce sheath length relative to wt control (Fig. 4h), indicating that length reduction is specific to disruption of PDZ binding for each of these proteins. Oligodendrocytes expressing full-length proteins also have normal number of sheaths (Fig. 4i) and normal total membrane (Fig. 4j). These data illustrate that transsynaptic adhesion molecules, typically studied only in the context of synapses, also determine the length and number of myelin sheaths formed by oligodendrocytes.

**Cadm1b functions at the axon-myelin junction to promote ensheathment.** Our dominant-negative approach tested the requirement for each candidate's PDZ binding motif in modulating myelination. PDZ binding is essential for downstream signaling through postsynaptic scaffolds, including PSD95 and CASK, in the recipient cell (or sheath). To determine where in the oligodendrocytes these candidates act to modulate ensheathment, we focused on one candidate, Cadm1b. We first confirmed expression of *cadm1b* in oligodendrocytes by generating a knock-in transgenic, using the same CRISPR/Cas9-mediated enhancer trap strategy as for *dlg4b* (Fig. 3a). Nearly all *mbpa*:tagRFP + oligodendrocytes are also *cadm1b* + but not all *cadm1b* + cells are *mbpa*:tagRFP+, likely indicating neuronal expression (Fig. 5a).

To track Cadm1b localization we generated a fusion protein, eGFP-Cadm1b, and drove expression with *myrf* regulatory DNA. eGFP-Cadm1b is distributed in both the somatic and sheath compartments (Figs. 5b, b′). In sheaths, eGFP-Cadm1b signal resembles two parallel lines demarcating the sheath perimeter, raising the possibility that eGFP-Cadm1b is transmembrane in sheaths. To estimate the fraction of our fusion protein associated with sheath membrane vs cytosol, we expressed eGFP-Cadm1b in oligodendrocytes that were transgenically co-labeled with either *Tg(sox10:tagRFP)*, which expresses a cytosolic RFP (Fig. 5b), or *Tg(sox10:mRFP)*, which expresses a membrane-tethered RFP (Fig. 5b′). We calculated Mander's colocalization coefficient[30] and found that eGFP-Cadm1b signal robustly colocalizes with

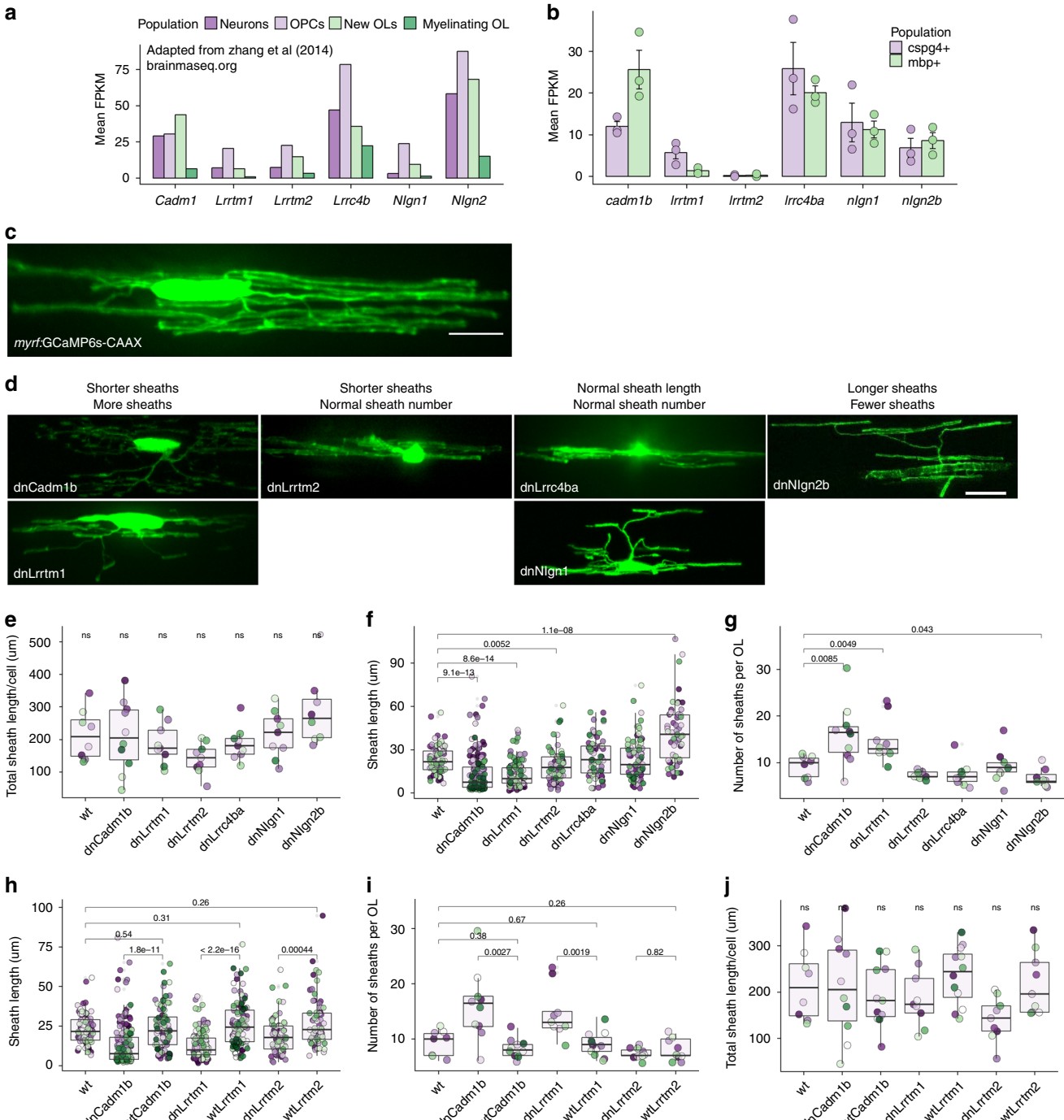

**Fig. 4** Candidate synaptogenic adhesion molecules have variable effects on myelin sheath length and number. **a** RNA-seq FPKM values for mouse cortical neurons, OPCs, new oligodendrocytes, and myelinating oligodendrocytes for candidates Cadm1, Lrrtm1, Lrrtm2, Lrrc4b, Nlgn1, and Nlgn2. Plot generated using publicly available values from brainrnaseq.org[14]. **b** RNA-seq FPKM values for FAC-sorted zebrafish *olig2 + /cspg4 +* (cspg4 +) and *mbpa + /olig2 +* (mbp +) cells for zebrafish homologs of the same candidates[36]. **c** Oligodendrocyte transiently expressing *myrf*:GCaMP6s-CAAX. Scale bar 10 μm. **d** Candidate dominant-negative (dn) alleles expressed transiently as *myrf*:dnX-2A-GCaMP6s-CAAX and grouped by effect on sheath length and number. Scale bar 10 μm. **e** Total sheath length per cell expressing each dominant-negative candidate was unchanged by all candidates, Kruskal–Wallis test. Dot color marks individual cells and matches colored dots in corresponding sheath length and number plots. **f** Sheath length values for each dominant-negative candidate. *n* (cells/sheaths) = 8/74 wt, 10/159 dnCadm1b, 9/132 dnLrrtm1, 9/67 dnLrrtm2, 9/68 dnLrrc4ba, 9/83 dnNlgn1, 8/55 dnNlgn2b. Data in plots **f–i** were tested by Wilcox rank-sum with Bonferroni-Holm correction for multiple comparisons. dnLrrc4ba/wt and dnNlgn1/wt comparisons were ns. **g** Sheath number values for each dominant-negative candidate (*n* listed in **f**). **h** Decreased sheath length in dnCadm1b, dnLrrtm1, and dnLrrtm2 cells is specific to the DN disruption, because expression of the full-length (wt) protein does not reduce sheath length. *n*(cells/sheaths) = 11/91 wtCadm1b, 12/108 wtLrrtm1, 9/72 wtLrrtm2. **i** Expression of WT forms of the three candidates that reduced sheath length resulted in normal sheath number. **j** Total sheath length per cell expressing each WT candidate was not significantly different for any group, Kruskal–Wallis test

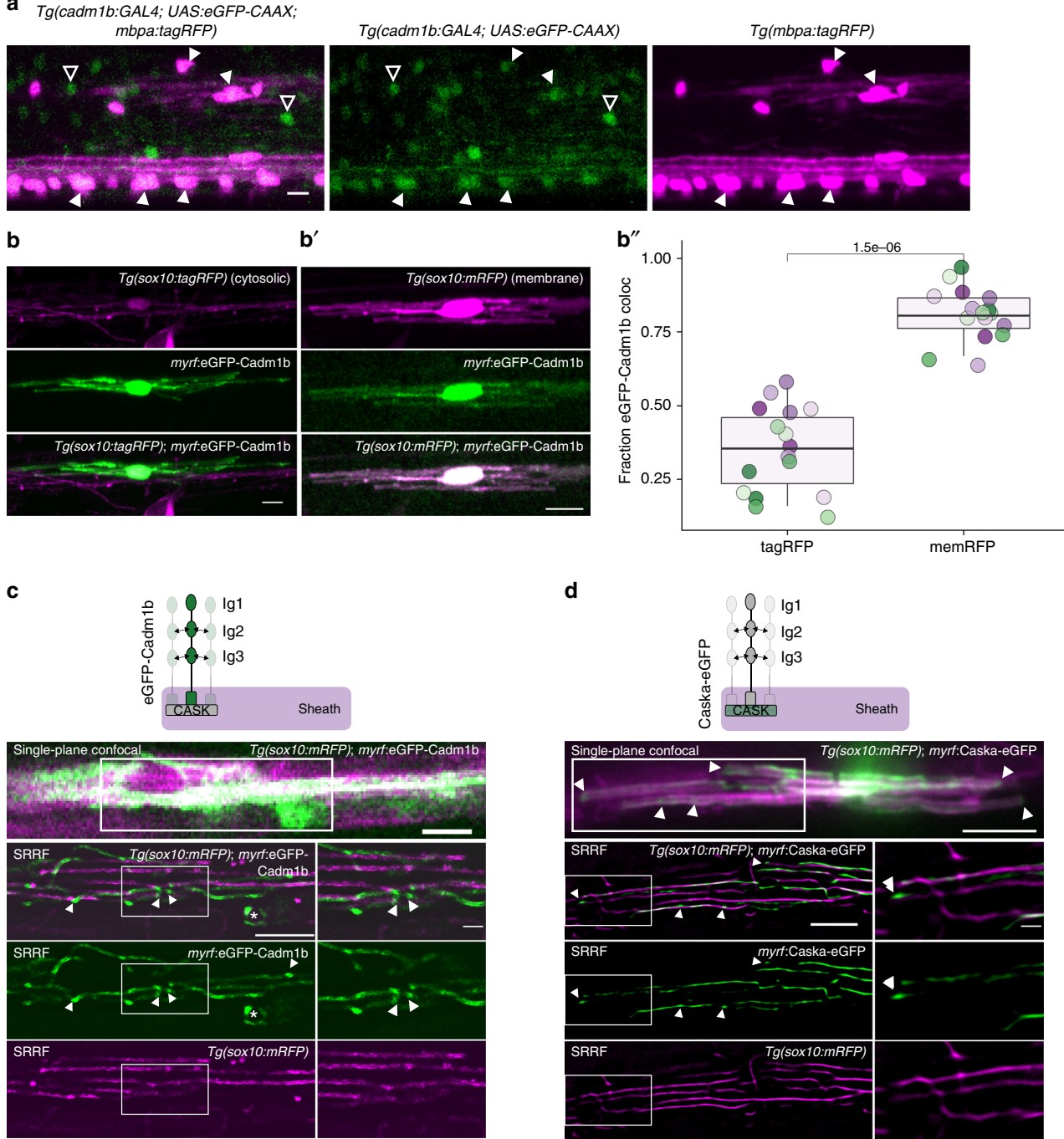

**Fig. 5** Cadm1b localizes to myelin sheath membrane. **a** CRISPR/Cas9-mediated GAL4 enhancer trap of *cadm1b* reports cells expressing *cadm1b* via *UAS:* eGFP-CAAX expression. Larvae additionally carrying *Tg(mbpa:tagRFP)* reveal *mbp* + myelinating oligodendrocytes. Closed arrowheads indicate *mbp* +, *cadm1b* + oligodendrocytes, whereas open arrowheads indicate *mbp*-, *cadm1b* + cells (likely neurons). Scale bar, 10 μm. **b**, **b'** Expression of *myrf*:eGFP-Cadm1b in *Tg(sox10:tagRFP)* (**b**) or *Tg(sox10:mRFP)* (**c**) co-labeled oligodendrocytes. **b''** Fraction of eGFP-Cadm1b colocalized with RFP (Mander's colocalization coefficient) for *Tg(sox10:tagRFP)* (tagRFP) and *Tg(sox10:mRFP)* (memRFP). *n* = 16 cells (tagRFP), 16 cells (memRFP), Wilcox rank-sum test. Colored dots represent individual cells. **c** Confocal single-plane (top) and super resolution radial fluctuations (SRRF) (bottom 3 panels) imaging of a eGFP-Cadm1b expressing oligodendrocyte in a *Tg(sox10:mRFP)* larva. Arrowheads indicate eGFP-Cadm1b puncta that are not present in mRFP SRRF. Asterisk marks a sheath going in to the plane of view with circular edges surrounded by eGFP-Cadm1b puncta. Scale bars 5 μm except for innermost inset, 1 μm. **d** Similar to **c** for an oligodendrocyte expressing Caska-eGFP. Scale bars 10 μm (top), 5 μm (left SRRF), 1 μm (SRRF insets)

membrane RFP but only minimally with cytosolic RFP (Fig. 5b"), indicating that eGFP-Cadm1b is membrane-localized in sheaths.

To resolve eGFP-Cadm1b localization in sheaths, we used super resolution radial fluctuations imaging (SRRF)[39]. We used Fourier ring correlation analysis to determine that SRRF yields resolution of ~120 nm in cells in live zebrafish (Supplementary Fig. 2). SRRF reveals eGFP-Cadm1b puncta primarily at the terminal ends of sheaths (Fig. 5c). To determine whether SRRF would report this labeling pattern for any membrane-associated protein, we performed SRRF imaging of cells expressing both

*sox10*:mRFP and *myrf*:eGFP-Cadm1b (Fig. 5c). Most eGFP-Cadm1b puncta are not associated with mRFP puncta, suggesting that eGFP-Cadm1b enrichment at sheath ends is specific to Cadm1b rather than a general feature of membrane-associated proteins. We similarly analyzed the PDZ scaffold that anchors Cadm1b, Caska/CASK[33]. Caska-eGFP occupies a similar position as eGFP-Cadm1b at the ends of sheaths (Fig. 5d). These data suggest that the synaptic features of the axon-myelin interface are primarily present at the terminal ends of sheaths, a location that also harbors axon-myelin adhesive complexes including glial NF155 binding to axonal Caspr and Contactin[40].

Sheath membrane localization is consistent with Cadm1b functioning as a transmembrane protein, possibly bridging the periaxonal space to interact with an axonal partner. Although Cadm proteins can interact transsynaptically both homo- and heterophilically, Cadm1 binds most strongly to Cadm2[41], a protein previously found to exclusively label myelinated axons in the CNS and to adhere to an unknown partner in oligodendrocytes[42]. To test the possibility that oligodendrocyte Cadm1b interacts with axonal Cadm2, we generated eGFP-Cadm2a and drove expression in reticulospinal neurons. eGFP-Cadm2a localizes under sheaths (Fig. 6a) raising the possibility that Cadm2a and Cadm1b interact across the axon-myelin interface. However, this does not exclude the possibility that Cadm1b interacts with other axonal Cadm partners. To test whether Cadm1b interacts with an axonal partner to drive myelination, we generated a second dominant-negative allele designed to prevent adhesion to transsynaptic partners. This allele, Ig1-dnCadm1b, lacks the extracellular distalmost Ig-like domain (Ig1) that specifically interacts transsynaptically with Cadm partners located on other cells[41]. We excised Ig1 while preserving the rest of the extracellular domain because the proximal Ig-like domains (Ig2, Ig3) allow Cadm1b to be incorporated into *cis* oligomers with other Cadm1b molecules. In this way, Ig1-dnCadm1b is predicted to bind endogenous Cadm1b via lateral Ig2 and Ig3 interactions, but to interfere with adhesion in *trans*[41] (Fig. 6b).

Expression of Ig1-dnCadm1b in oligodendrocytes produces a phenotype distinct from expression of the wt form or the PDZ binding dominant-negative (hereafter called PDZIIb-dnCadm1b) (Fig. 6c). Ig1-dnCadm1b oligodendrocytes generate sheaths that are significantly shorter than wt and wtCadm1b-expressing oligodendrocytes but longer than those made by PDZIIb-dnCadm1b oligodendrocytes (Fig. 6d). Furthermore, the number of sheaths is modestly increased, but not significantly different than wt or wtCadm1b sheath number (Fig. 6e). The myelinating capacity of cells expressing each of the constructs is unchanged (Fig. 6f), indicating that both extracellular adhesion and PDZ binding tune how oligodendrocytes distribute myelin among sheaths rather than influencing myelin production.

Because PDZIIb-dnCadm1b specifically disrupts downstream Cadm1b interactions with scaffolds including CASK, syntenin, and Mpp3[33,43], whereas Ig1-dnCadm1b disrupts extracellular adhesion without changing PDZ interactions[41], these data suggest that Cadm1b has roles in both "postsynaptic" signaling within sheaths as well as transsynaptic adhesion-induced signaling with the axon to tune ensheathment. At synapses, the Ig1 domain of postsynaptic Cadm1 functions both to adhere the pre- and postsynapse and to induce presynaptic assembly in the presynaptic neuron[33].

Does the Ig1 domain of Cadm1b function to induce presynaptic vesicle clustering in the axon? We first tested whether oligodendrocyte Cadm1b apposes neuronal synaptic vesicles by expressing Syp-mScarlet pan-neuronally in axons and eGFP-Cadm1b in oligodendrocytes, and found that Syp-mScarlet puncta cluster near sites of eGFP-Cadm1b signal (Fig. 6g). Because synaptic vesicle exocytosis promotes sheath growth[11–13],

we next tested the possibility that Ig1-dnCadm1b oligodendrocytes form shorter sheaths due to impaired vesicle clustering. We labeled either wt or Ig1-dnCadm1b oligodendrocytes in pan-neuronal Syp-mScarlet larvae and observed axonal Syp-mScarlet wrapped by myelin sheaths (Fig. 6h, h′). Unlike the sparse labeling of *phox2b* + neuronal synaptic vesicles (Fig. 1c), this pan-neuronal labeling allowed us to examine all synaptic vesicles but precluded us from measuring vesicle signal intensity in individual sheaths due to surrounding labeled vesicles. We instead employed Mander's colocalization to assess the 3D fraction of oligodendrocyte-labeled territory that was also positive for Syp-mScarlet signal at 4 dpf, when vesicle clustering is evident (Fig. 1d). Colocalization values for sheaths expressing Ig1-dnCadm1b are more variable than values for control sheaths but differences between the two groups are not statistically significant using the standard of $p < 0.05$ (Fig. 6i). This result suggests that disruption of Cadm1b function alters sheath growth independently of presynaptic assembly. Importantly, Ig1-dnCadm1b oligodendrocytes continue to exhibit abnormal ensheathment later in development (Fig. 6j, k, l), indicating that compromised axon-myelin adhesion blocks, rather than delays, sheath growth.

## Discussion

The data and conclusions that we present in this manuscript add to previous evidence of synaptic-like communication between neurons and oligodendrocyte lineage cells. Oligodendrocyte precursor cells (OPCs) that express the proteoglycan NG2 receive *bona fide* synaptic input from pyramidal neurons and interneurons in hippocampus[44,45], axons in corpus callosum[46], and climbing fibers in cerebellum[47]. However, neurotransmitter ceases to evoke somatic currents in oligodendrocytes following their differentiation from NG2 glia[46,48], suggesting that synapses on to OPCs are disassembled upon myelination. Recently, live-imaging revealed neuronal activity-evoked calcium transients in nascent sheaths[49,50], indicating that neuronal activity continues to elicit local ion flux in oligodendrocyte sheaths after somatic currents are no longer detectable. In the optic nerve, axo-myelinic synapses mediated by axonal glutamate release onto myelinic NMDA receptors might allow axons to dynamically control metabolic support provided by overlaying myelin[51,52]. Potentially, synaptic-like communication between axons and oligodendrocyte lineage cells at distinct stages of differentiation mediates distinct features of myelin plasticity and function, such as OPC number, choice of axons for ensheathment, myelin coverage of axons, and metabolic transfer.

Despite the evidence for synaptic-like communication between axons and oligodendrocyte lineage cells, the molecular mechanisms that mediate it have not been determined. If axons communicate to oligodendrocytes via secreted neurotransmitters and trophic factors, then oligodendrocytes might utilize canonical postsynaptic molecular assemblies to receive it. Oligodendrocytes express the postsynaptic scaffold PSD95, shown by immunoblot[53] and RNA-seq[14–16], but the subcellular localization of the protein was not known. We used a genetically-encoded intrabody and an antibody to label endogenous PSD95 in oligodendrocytes and discovered heterogeneity in the labeling of sheaths of individual oligodendrocytes. Some sheaths exhibited end labeling only, while others exhibited a periodic spread of puncta along the length of the sheath. Still other sheaths exhibited no defined labeling. Why this diversity? Perhaps sheaths, like dendritic spines, use different scaffolds to anchor unique subsets of receptors and signaling molecules required for autonomous interaction with the presynaptic axon. For example, excitatory synapses use PSD95 to anchor NMDA receptors, whereas inhibitory synapses primarily use Gephryin (Gphn) to anchor GABA receptors[54].

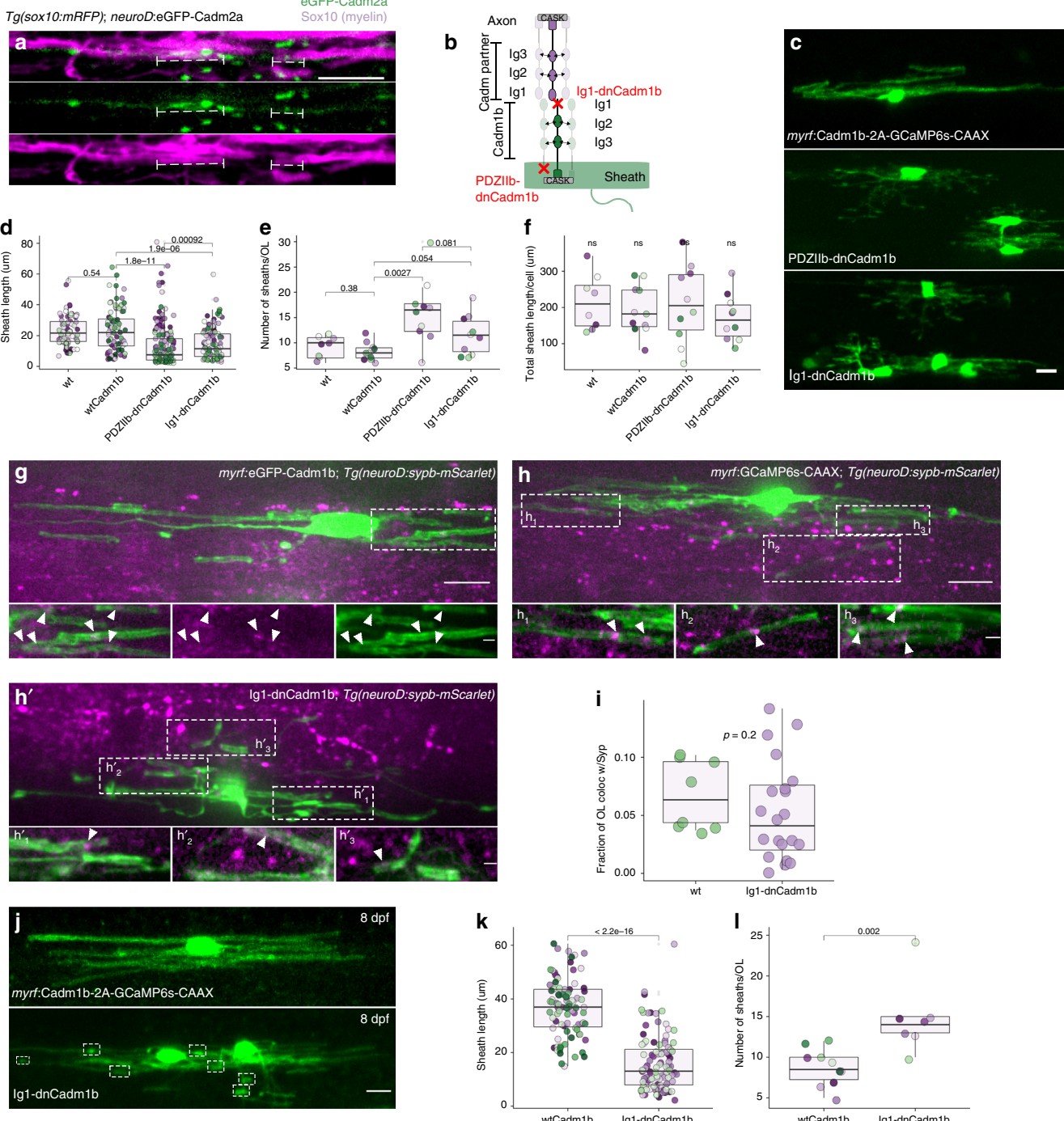

Oligodendrocytes also express Gphn and other scaffolds[14], raising the possibility that sheaths labeled poorly for PSD95 might instead utilize other protein scaffolds, perhaps to support sheaths on different classes of neurons. We also found diverse patterns of vesicle distribution within myelinated axons, including repeated examples in which vesicles appeared clustered near the ends of overlying nascent myelin sheaths, distributed along the length of axons coincident with overlying sheaths or distributed uniformly along both ensheathed and bare portions of axons. Although we do not know the functional significance of these distinct localization patterns, they might reflect distinctions in neuron class, neurotransmitter type or mechanism of signal release.

The presence of these canonical synaptic structures at the axon-myelin interface is consistent with the possibility that these proteins function similarly as they do at neuronal synapses, participating in exocytosis of neurotransmitter and anchoring of receptors to support communication. However, other functions are also possible for these features. For example, "synaptic" vesicles are identified, by us and by others, by the presence of membrane proteins such as Synaptophysin (Syp) that are generally absent in non-synaptic vesicles. A limitation of this classification is that Syp presence in a vesicle membrane indicates nothing about the cargo contained. It is possible that the Syp-labeled vesicles docked under myelin represent a distinct pool

**Fig. 6** The extracellular, trans-acting adhesion domain (Ig1) of Cadm1b promotes myelin sheath growth. **a** Neuronal eGFP-Cadm2a puncta (green) localize under myelin sheaths (magenta, brackets). Scale bar, 10 μm. **b** Schematic of dominant-negative oligodendrocyte Cadm1b variants, PDZIIb-dnCadm1b and Ig1-dnCadm1b, interacting with partner Cadm located on neurons. **c** Representative examples of oligodendrocytes expressing wildtype Cadm1b, PDZIIb-dnCadm1b, and Ig1-dnCadm1b with GCaMP6s-CAAX to label sheath membrane. Scale bar, 10 μm. **d** Sheath lengths for wt, wtCadm1b-, PDZIIb-dnCadm1b-, and Ig1-dnCadm1b-expressing oligodendrocytes. Note that PDZIIb-dnCadm1b is the same allele presented in Fig. 4. *n* (cells/sheaths) = 8/74 (wt), 11/91 (wtCadm1b), 10/159 (PDZIIb-dnCadm1b), 10/115 (Ig1-dnCadm1b), Wilcox rank-sum test with Bonferroni-Holm correction for multiple comparisons. Dots of the same color indicate sheaths belonging to the same cell and match those in plots **e** and **f**. **e** Sheath number for wt, wtCadm1b-, PDZIIb-dnCadm1b-, and Ig1-dnCadm1b-expressing oligodendrocytes. Same n and statistical test as in **d**. **f** Total sheath length generated per cell is unchanged by all alleles, Kruskal–Wallis test. (**g**, **h**, **h′**) Max projection image of *myrf*:eGFP-Cadm1b oligodendrocyte in *Tg(neuroD:sypb-mScarlet)* larvae, in which Syp-mScarlet is expressed pan-neuronally. Inset is a substack projection of 6 slices (0.31 μm/slice) containing eGFP-Cadm1b-labeled sheaths wrapping Syp-mScarlet puncta. Arrowheads indicate colocalized oligodendrocyte eGFP-Cadm1b and neuronal Syp-mScarlet signal. **h**, **h′** *myrf*:GCaMP6s-CAAX (wt) (**h**) or Ig1-dnCadm1b (**h′**) oligodendrocytes in *Tg(neuroD:sypb-mScarlet)* larvae. Insets are substack projections of 2–8 slices containing sheaths wrapping Syp-mScarlet puncta. In **g**, **h′** scale bars are 10 μm/2 μm in insets. **i** Mander's colocalization coefficient displaying the fraction of GCaMP6s-CAAX oligodendrocyte signal that is positive for Syp-mScarlet signal in wt and Ig1-dnCadm1b cells in 4 dpf larvae. *n* = 8 cells (wt) and *n* = 20 cells (Ig1-dnCadm1b), Wilcox rank-sum test. **j**, **k**, **l** By 8 dpf, Ig1-dnCadm1b-expressing oligodendrocytes still exhibit numerous stunted myelin sheaths (boxes) compared to wildtype Cadm1b-expressing cells. Sheath length (**k**) and number (**l**) for *n* (cells/sheaths) at 8 dpf = 10/87 (wtCadm1b) and 7/104 (Ig1-dnCadm1b), Wilcox rank-sum test. Scale bar, 10 μm

from the recycling pool that is shared among presynaptic boutons[55], and may contain distinct cargo intended for myelin, such as pro-myelinating factors. A similar argument applies to our observation that canonical postsynaptic scaffolds, including PSD95 and CASK, are present in myelin sheaths. While these scaffolds could be anchoring receptors and signaling molecules as they do at synapses, myelinating oligodendrocytes downregulate expression of many receptors, raising the possibility that these scaffolds anchor other proteins or serve other functions. One functional similarity that we did observe is that of the synaptogenic adhesion molecules: interruption of these proteins in neurons disrupts synapse size and number, and we observed a reduction in myelin sheath length. These proteins are sufficient for presynaptic assembly when expressed in non-neuronal cells. However, we observed a severe reduction in myelin sheath length upon expression of dominant-negative that did not diminish presynaptic vesicle clustering, indicating that adhesion may be a more important function for these molecules at the axon-myelin interface.

Why have presynaptic and postsynaptic machinery not been discovered at the axon-myelin interface by transmission electron microscopy (EM), a technique that has been used to identify synapses for over 30 years? One possibility is that EM identifies mature synapses, but frequently misses small, nascent, and diverse synapses that do not resemble classic EM synapses[56]. Wake et al. (2015) used EM to assess axon-OPC contacts in vitro. While they observed presynaptic vesicles docked at axon-OPC contacts, they did not identify a postsynaptic density (PSD), which led them to conclude that the junction is non-synaptic[13]. In contrast, by using a non-EM approach, we have detected PSD95 in sheaths. What explains this difference? Perhaps axon-OPC junctions have PSDs on par with immature synapses, which lack an EM-resolvable PSD until maturity, or perhaps maturity of axon-OPC junctions was not achieved in culture. Another possible reason that synaptic features of axon-oligodendrocyte contacts may go unnoticed by EM is diversity: perhaps synaptic features are not equally present at all axon-oligodendrocyte junctions. Like Wake et al. (2015), Doyle et al. (2017) identified docked axonal vesicles under myelin, but not all myelinated axons had sub-myelin docked vesicles[57]. This is consistent with the observation that vesicular release from certain axons profoundly modulates their myelin profiles, whereas vesicular release is dispensable for the myelination of other classes of axons[58]. Furthermore, while we found PSD95 localization in most sheaths, there was substantial heterogeneity between sheaths of individual cells. Together, these studies raise the possibility that axon-oligodendrocyte contacts are diverse in their usage of synaptic

elements, perhaps as broad in scope as the variety of synapses. This prompts a significant need for alternatives to EM to identify diverse synapses. Burette et al.[56] set out recommendations for single-synapse analysis for those synapses missed by EM, including fluorescence microscopy and optophysiology approaches, both of which we have used here.

To test whether these synaptic similarities are integral to normal myelination, we manipulated synaptic adhesion proteins in oligodendrocytes that disrupt synapse formation in neurons. We tested dominant-negative alleles of candidate transsynaptic adhesion molecules expressed by oligodendrocytes and measured oligodendrocyte sheath length, number, and total sheath length generated per cell. We discovered specific requirements for PDZ binding of Cadm1b, Lrrtm1, and Lrrtm2 in oligodendrocyte sheath length. Lrrtm1 and Lrrtm2 bind to the PDZ domains of PSD95[34], but Cadm1b is instead anchored by the related scaffold CASK, which also anchors the major oligodendrocyte protein Claudin11[59]. Like PSD95, we found that Cadm1b and its interacting scaffold Caska appear to be concentrated near the ends of nascent myelin sheaths during development. This raises the possibility that the ends of sheaths are sites of axonal signal reception and intracellular signal transduction that promote sheath growth (Fig. 7). Consistent with this possibility, oligodendrocyte expression of a form of Cadm1b lacking the intracellular PDZ domain, which mediates intracellular protein interactions, produced abnormally short sheaths. Oligodendrocyte expression of a different Cadm1b variant designed to interfere with binding to transsynaptic adhesion partners also produced shorter myelin sheaths, but the change was less extreme than that caused by the PDZ deletion variant. Thus, Cadm1b might have distinct signaling and adhesion-based functions in myelin sheath growth. Although we did not find evidence that disruption of Cadm1b function alters presynaptic assembly, as manifested by axonal vesicle clustering, multiple transsynaptic adhesion systems are thought to operate in parallel at synapses[60,61], and redundancy within and between families can obscure how individual adhesion proteins contribute to synapse formation. Intriguingly, oligodendrocyte expression of a Cadm4 variant, a protein with structural similarity to Cadm1b, consisting of the extracellular and transmembrane domains, predicted to promote adhesion, resulted in a large excess of short sheaths[62]. Thus, axon-oligodendrocyte interactions that promote sheath growth might be mediated by numerous adhesion molecules.

We have shown that synaptic features are present at the axon-myelin junction and that synaptogenic adhesion molecules have a previously underappreciated role in shaping myelination. Is this similarity to synaptogenesis biologically significant? Dendrites

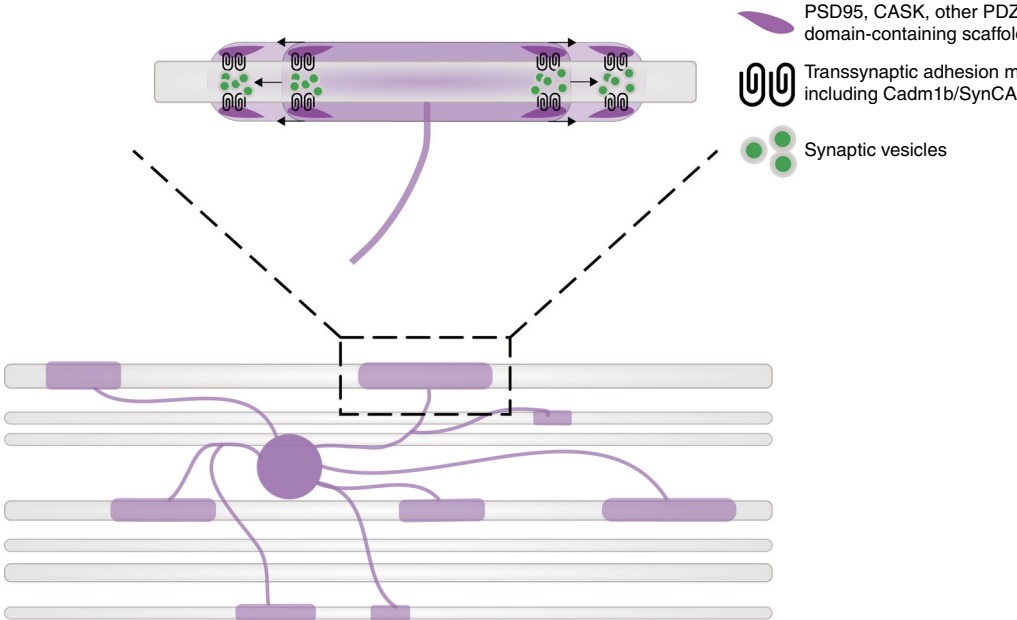

**Fig. 7** Working model of adhesion-promoted sheath growth. PSD95 and other PDZ domain-containing scaffolds expressed by oligodendrocytes, including CASK and Dlg1, anchor transsynaptic adhesion molecules that adhere to and induce presynaptic assembly in the ensheathed axon. The accumulation and exocytosis of synaptic vesicles at ensheathment sites promotes the elongation of nascent sheaths, possibly via neurotransmitter- and/or neurotrophin-induced intracellular signaling in the sheath

and oligodendrocytes utilizing similar mechanisms to establish stable contacts with axons raises the possibility that oligodendrocytes may be vulnerable to pathology in psychiatric disease, where mutations in synaptic genes are likely drivers of pathogenesis[63]. Indeed, ultrastructural analysis of postmortem brain tissue from patients with schizophrenia reveals myelin and oligodendrocyte defects[64] and diffusion tensor imaging of patients living with schizophrenia[65] and autism[66] reveal white matter defects. Together, these findings raise the intriguing possibility that mutations in synaptic genes could affect oligodendrocytes cell-autonomously to disrupt myelination, which may then contribute to disease progression via reduced neuronal support or conduction velocity. Alternatively, mutations in synaptic genes could impair synaptic transmission and thus change neuronal activity in a way that disrupts oligodendrocyte maturation, myelination, or survival. We have shown that a number of synaptic proteins are expressed at the axon-myelin interface and that disruption of some of these proteins in oligodendrocytes disrupts myelination. This finding supports the former possibility, that synaptic genes cell-autonomously regulate myelination in oligodendrocytes. However, because the oligodendrocyte lineage is sensitive to perturbations in neuronal activity[9], it is also possible that reduced myelination resulting from synaptic protein dysfunction in oligodendrocytes causes aberrant or asynchronous neuronal activity which then further disrupts the oligodendrocyte lineage. Untangling the relationship between neuronal activity and oligodendrocyte myelination is crucial for understanding nervous system function in both health and disease. Our work draws structural and functional parallels between synaptogenesis and myelination and provides new targets for investigating the mechanistic basis of developmental myelination.

## Methods

**Zebrafish lines and husbandry**. All animal work was approved by the Institutional Animal Care and Use Committee at the University of Colorado School of Medicine. Zebrafish embryos were raised at 28.5 °C in embryo medium and staged according to hours or days post-fertilization (hpf/dpf) and morphological criteria. Tg(sox10:mRFP)[vu234], Tg(sox10:tagRFP)[co26], Tg(phox2bb:GAL4)[co21], Tg

(UAS:syp-eGFP)[67], Tg(mbpa:tagRFP)[co25], Tg(mbpa:eGFP-CAAX), and Tg(UAS:eGFP-CAAX)[co18] transgenic lines were used. In addition, we generated and used the new lines Tg(cadm1b:GAL4)[co53], Tg(dlg4b:GAL4)[co54] (see section on enhancer trapping), and Tg(neuroD:sypb-mScarlet)[co55]. All other reporters were expressed by transient transgenesis to achieve sparse labeling for single cell analysis.

**Plasmid construction and generation of transgenic zebrafish**. Tol2 expression plasmids were generated by Multisite Gateway cloning and injected into 1-cell embryos with Tol2 mRNA to generate transient-transgenic animals. The following entry clones were used (cited) or made (see Supplementary Table 1) via BP recombination of appropriate pDONR backbone with attB-flanked regulatory elements or coding sequences.

p5E-neuroD, p5E-4xUAS[68], p5E-myrf (gift from Jacob Hines)

pME-eGFP[68], pME-GAL4[68], pME-vamp2, pME-sypHy, pME-GCaMP6s-CAAX, pME-cadm1b, pME-Ig1-dnCadm1b, pME-PDZIIb-dnCadm1b, pME-lrrtm1, pME-dnLrrtm1, pME-lrrtm2, pME-dnLrrtm2, pME-dnLrrc4ba, pME-dnNlgn1, pME-dnNlgn2b, pME-PSD95.FingR-GFP-CCR5TC-KRAB(A), pME-caska, pME-sypb

p3E-polyA[68], p3E-eGFP[68], p3E-2A-GCaMP6s-CAAX, p3E-cadm1b, p3E-cadm2a, p3E-2A-BoNT/B, p3E-2A-dnVamp2, p3E-mScarlet

p3E-2A-GCaMP6s-CAAX, p3E-2A-BoNT/B, and p3E-2A-dnVamp2 were made by amplifying restriction enzyme site-flanked inserts and subcloning into digested p3E-2A-mcs (see table of primers).

Entry clones were LR-recombined with either of destination vectors pDEST-Tol2-CG2 (green heart marker) or pDEST-Tol2-CC2 (blue eye marker) as transgenesis indicators[68].

zcUAS:PSD95.FingR-GFP-CCR5TC-KRAB(A) was a gift from Joshua Bonkowsky (Addgene plasmid #72638) and was used to generate pME-PSD95.FingR-GFP-CCR5TC-KRAB(A). CMV::SypHy A4 was a gift from Leon Lagnado (Addgene plasmid #24478) and used to generate pME-sypHy. pGP-CMV-GCaMP6s was a gift from Douglas Kim (Addgene plasmid #40753) and used to generate pME-GCaMP6s-CAAX and p3E-2A-GCaMP6s-CAAX. 14UAS:zfPSD95:GFP was a gift from Martin Meyer & Stephen Smith (Addgene plasmid #74314). pQL86-eGFP-BoNT/B was a gift from Chandra Tucker and used to generate p3E-2A-BoNT/B. pmScarlet_C1 was a gift from Dorus Gadella (Addgene plasmid #85042) and used to generate p3E-mScarlet.

**CRISPR/Cas9-mediated enhancer trapping**. We knocked in GAL4 200–500 bp upstream of the translation start site of cadm1b and dlg4b using a previously published method[25]. Briefly, mbait-hs-Gal4 plasmid, mBait guide RNA, guide RNA targeting upstream of our genes of interest (cadm1b, chr15:18645374-18645393; dlg4b, chr23:44280085-44280104 in GRCz11/danRer11 assembly, see primer table for sequences), and Cas9 mRNA were injected into 1-cell Tg(UAS:eGFP-CAAX) embryos. Guide RNAs were designed using crispor.tefor.net[69] with specificity >99. F0 founders were screened for expression and raised to adulthood. Experiments

involving these knock-ins were only performed on stable transgenic (F1 or later) *Tg (cadm1b:GAL4)*[co53] and *Tg(dlg4b:GAL4)*[co54] larvae.

**Dominant negative allele generation**. We made zebrafish homologs of published mouse dominant-negative alleles for all candidates. For candidates with duplicate paralogs in the zebrafish genome, we investigated the paralog with higher expression in oligodendrocytes[36]. We isolated RNA from whole, wildtype AB-strain 4 dpf larvae and synthesized cDNA (iScript) to use as template to high-fidelity (Phusion) PCR amplify attB-flanked coding sequences for candidates, omitting C-terminal PDZ binding sites for Cadm1b (-KEYYI)[38,70], Lrrtm1 and Lrrtm2 (-ECEV)[34], Nlgn1 and Nlgn2b (-STTRV)[37], and Lrrc4ba (-ETQI)[34]. For Lrrtm1/2, we omitted 51 additional C-terminal residues to make alleles similar to those used previously[34]. Full-length sequences were also amplified. AttB-flanked products were recombined with *pDONR-221* to form middle entry vectors, and subsequently recombined with *p5E-myrf*, *p3E-2A-GCaMP6s-CAAX*, and *pDEST-Tol2-CG2* to generate expression constructs.

**Immunohistochemistry**. 5 dpf *Tg(mbpa:eGFP-CAAX)* larvae injected at one cell stage with *p.neuroD:sypb-mScarlet* were fixed (4% paraformaldehyde/1X PBS) overnight at 4 °C. After rinsing in PBS, larvae were embedded in 1.5% agar/30% sucrose blocks and then soaked in 30% sucrose overnight at 4 °C. Next, the blocks were frozen in 2-methyl butane chilled by immersion in liquid nitrogen. 20 μm sections were cut in sagittal plane and collected on microscope slides. The sections were rehydrated with PBS and then blocked with AB Blocking Solution (2% goat serum/1% bovine serum albumin/1% DMSO/0.25% Triton X-100) for 1 h at room temperature. Rabbit anti-PSD95 antibody (abcam, ab18258) was applied at 1:1000 dilution in AB Blocking Solution for 20 h at 4 °C. The sections were washed extensively with PBS and then incubated with Alexa Fluor 647 goat anti-rabbit antibody (Jackson ImmunoResearch, 111-605-003) at 1:200 dilution in AB Blocking Solution for 20 h at 4 °C. After extensive washing with PBS, the slides were covered with Vectashield and coverslipped. Images were collected using a Zeiss CellObserver Spinning Disk confocal system, C-Apochromat 63×/1.20 NA water immersion objective and Photometrics Prime 95B camera.

**Imaging and image analysis**. We performed live-imaging on larvae embedded laterally in 1.2% low-melt agarose containing either 0.4% tricaine or 0.3 mg/ml pancuronium bromide for immobilization. We acquired images on a Zeiss Axiovert 200 microscope with a PerkinElmer spinning disk confocal system (PerkinElmer Improvision), a Zeiss LSM 880 (Carl Zeiss), or a Zeiss CellObserver Spinning Disk confocal system, C-Apochromat 63×/1.20 NA water immersion objective and Photometrics Prime 95B camera. For Figs. 1 and 2, we collected data from one field of view from each animal. For experiments in Figs. 3–6 we only used one cell per animal, so n = number of cells and number of fish. After collecting images with Volocity (PerkinElmer) or Zen (Carl Zeiss), we performed all processing and analysis using Fiji/ImageJ. Image analysis was performed blind by using the Fiji plugin Lab-utility-plugins/blind-files.

**Synaptophysin-eGFP clustering**. We imaged *Tg(phox2b:GAL4; UAS:syp-eGFP; sox10:mRFP)* larvae at 3, 4, and 5 dpf. Roughly 50% of *phox2b*+ neurons are myelinated[11], so we confirmed myelination status of individual *phox2b*+ neurons by taking oversampled z-stacks and examining orthogonal views. We traced individual *phox2b*+ neurons containing Syp-eGFP puncta with Fiji's Plot Profile tool and counted peaks, representing individual puncta, that exceeded 150% of the inter-peak background fluorescence intensity. The number of peaks (puncta) per μm of ensheathed and bare axon were transferred from Fiji to R and density of puncta in both regions was determined by dividing puncta count by length (for bare axon density, we used the length of axon present in the field of view). Regions were compared age-wise by Wilcox rank-sum test (wilcox.test, R package ggpubr) with no assumption of normality.

**SypHy signatures under myelin sheaths**. *neuroD*:sypHy; *Tg(sox10:mRFP)* larvae were immersed for 1 h in 1 μM bafilomycin A1 (Tocris, cat 1334) in DMSO vehicle in embryo medium. We then paralyzed embryos by adding pancuronium bromide (Sigma, cat P1918) to the solution to achieve a final concentration of 0.3 mg/ml. We made a small incision to the tip of the tail with a tungsten needle to promote circulation of the paralytic agent. Larvae were monitored for an additional 24 h post-experiment to ensure incisions did not cause lasting injury or fatality. The same bafilomycin/pancuronium bromide regimen was followed for larvae expressing *neuroD*:sypHy-2A-BoNT/B and *neuroD*:sypHy-2A-dnVamp2.

We imaged larvae in pancuronium bromide-containing low-melt agarose immersed in the same treated embryo medium (containing bafilomycin and pancuronium). We acquired oversampled z-stacks at a single x-y position, over the yolk extension, for individual larvae (1 z-stack per fish). Determination of whether an axonal sypHy hotspot was wrapped by mRFP + myelin was made by orthogonal views. Because category designation (punctate, filled, uniform) was subjective we plotted individual observations for transparency.

**PSD95 localization in sheaths**. We imaged transient-transgenic *myrf*:GAL4;UAS: PSD95-GFP larvae, *myrf*:PSD95.FingR-GFP-CCR5TC-KRAB(A); *Tg(sox10:mRFP)* larvae, and *myrf*:GAL4; *zcUAS*:PSD95.FingR-GFP-CCR5TC-KRAB(A); *Tg(sox10: mRFP)* larvae for examination of PSD95-GFP fusion protein localization as well as transcriptionally unregulated and regulated PSD95.FingR-GFP expression in oligodendrocytes. We designated categories based on transcriptionally-regulated PSD95.FingR-GFP expression. Because category designation (end, periodic, diffuse) was subjective, observations for individual cells were plotted for transparency. Unique cells were examined at each time point; i.e., cell #1 at 3 dpf is not cell #1 at 4 dpf. Sheaths overlapping in x-y space with the soma were frequently blown out so only those that did not overlap in x-y space with the soma were analyzed to prevent incorrect category assignment due to brightness. To test whether the distribution of categories changes over developmental time, we carried out a Chi square test (chisq.test, base R) on the contingency table for category at each age point.

**Dominant-negative tests and sheath measurements**. All imaging was performed at 4 dpf at approximately the same time each day (~100–102 hpf) or 8 dpf as a later time point. We predetermined that a sample size of 9–12 cells per candidate would generate a sufficient number of sheaths for both sheath length and number measurements. Sheaths were measured per cell by linear ROIs captured in ImageJ's ROI manager. Length values were exported to R with cell identification numbers to identify the number of lengths (sheaths) per cell. Total sheath length per cell is the sum of sheath lengths per cell identification number and sheath number is the number of lengths associated with an identification number. For all three parameters (total sheath length per cell, sheath length, and sheath number) we first tested for global significance using the Kruskal–Wallis test, and if this was significant, we made pairwise comparisons between groups using the Wilcox rank-sum test with p-values adjusted for multiple comparisons by the Bonferroni-Holm method.

**Quantitative colocalization**. We used the Fiji plugin JACoP (Just Another Colocalization Plugin)[30] to calculate Mander's Colocalization Coefficients, M1 and M2, which describe the fraction of protein A that colocalizes with protein B and vice versa, respectively[71]. We chose to evaluate colocalization on the basis of these coefficients because they are independent of fluorescence intensity, which varies between transient-transgenic cells and animals. To evaluate colocalization of eGFP-Cadm1b with RFPs, single-plane images of oligodendrocytes expressing *myrf*: eGFP-Cadm1b on either a membrane-labeling *Tg(sox10:mRFP)* or cytosol-labeling *Tg(sox10:tagRFP)* background were cropped to minimize background and split into separate channels for thresholding. JACoP generated M1 and M2 values, which were then exported to R for analysis. To evaluate colocalization of *myrf*:GCaMP6s-CAAX or *myrf*:Ig1-dnCadm1b-2A-GCaMP6-CAAX with *Tg(neuroD:sypb-mScarlet)*, z-stacks of 50 optical sections spaced 0.31 μm apart centered on individual oligodendrocytes were similarly blinded, cropped, thresholded, and M1 values analyzed in R.

**Super resolution radial fluctuations**. For individual cells, 100–200 single z-plane images were acquired with minimal time delay between images (acquisition rate ~5 frames per second). Images were transferred from Zen to Fiji and first corrected for drift (Plugins >Registration >Correct 3D drift). We corrected for slow drifts but did not edge-enhance images to avoid introducing artifacts. Drift-corrected images were then analyzed using the NanoJ-SRRF plugin with default settings.

**Statistics**. All statistics were performed in R (version 3.4.1) with RStudio. Plots were generated using dplyr and ggplot2 packages[72] with cowplot package formatting[73], and all statistical tests were performed using ggpubr[74] except for the $\chi^2$ test, which was carried out in base R[75]. We used the Wilcox rank-sum test (also called Mann–Whitney), with no assumption of normality, for all unpaired comparisons. For multiple comparisons, we first assessed global significance using the Kruskal-Wallis test, followed (only if Kruskal–Wallis significant) by pairwise Wilcox rank-sum tests with Bonferroni-Holm correction for multiple comparisons.

**Reporting summary**. Further information on research design is available in the Nature Research Reporting Summary linked to this article.

## Data availability
All data to support the conclusions of this study are included in the paper and Supplementary Material.

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

## Acknowledgements

We are grateful to Jacob Hines for cloning *myrf* regulatory DNA and providing it to us, to Christina Kearns for tissue sectioning, and to Dominik Stich for assistance with Fourier ring correlation. We also thank Ethan Hughes and Caleb Doll for valuable comments on the manuscript. This work was supported by US National Institutes of Health (NIH) grant R01 NS046668 and a gift from the Gates Frontiers Fund to B.A. and a National Science Foundation Graduate Research Fellowship (DGE-1553798) to A.N.H. The University of Colorado Anschutz Medical Campus Zebrafish Core Facility was supported by NIH grant P30 NS048154. All DNA plasmids and transgenic zebrafish used in this study are available by request.

## Author contributions

A.N.H. and B.A. conceived the project. A.N.H. performed all the experiments, with the exception of antibody labeling, which was performed by B.A., and collected and analyzed all the data. A.N.H. wrote and B.A. edited the manuscript.

## Additional information

**Competing interests:** The authors declare no competing interests.

