## [Peer Review File · Nature Communications]

Reviewers' comments:

Reviewer #1 (Remarks to the Author):

The authors have addressed my previous comments thoroughly in this interesting and important paper.

Reviewer #2 (Remarks to the Author):

The authors have done a satisfactory job addressing my technical comments on their manuscript. In their revision, the authors have added several controls (for example, BoNT/dnVamp2 control experiments) and validated their PSD-95 expression data (using orthogonal visualization techniques).

Overall this is a very interesting paper that will be an important contribution to the field and open up many doors for further scientific inquiry. My main reservation about the paper as written is that there is still no solid evidence that the oligodendrocyte-axon interaction is functionally similar to a neuronal synapse. The authors made several really intriguing observations that are consistent with this hypothesis, including (1) exocytosis along axon under myelin, (2) presence of PSD-95 in myelin sheaths, and (3) necessity of adhesion proteins that are also found in synapses for proper myelination. However there are several limitations to this "synaptic" interpretation of the data, including:

A) The authors present evidence for vesicular fusion occurring along axons. However, it is not clear what (if anything) is packaged inside these vesicles. Nor is there any data that examines functional relevance of these fusion events. If, for example, we could see that oligodendrocytes respond in some way to vesicular release along axons, this would better support the notion that axonal exocytosis is communicating to the oligodendrocyte (like a synapse). The authors note technical challenges in doing these types of experiments and say they are beyond the scope of the manuscript. Previous studies that blocked exocytosis and found myelination defects did not rule out alternative hypotheses, e.g. (1) release at synapse, not axon, signaling to myelin; or (2) exocytosis being important to deliver pro-myelination axonal membrane proteins rather than release of diffusible signals.

B) While the synaptic scaffolding protein localization data was validated using several methods, the functional relevance of PSD-95/CASK in myelination is still unknown. Again, are these proteins in OLs truly involved in setting up "synapse-like contacts?" What is the functional relevance of these proteins in the context of myelination? The authors make a good argument why this experiment is hard to do (e.g. two copies of PSD-95) but without functional insight it is hard to conclude much from these experiments apart from the fact that PSD-95 is found in myelin sheaths.

C) The paper provides evidence to support the hypothesis that there are adhesion proteins, that are also used at synapses, that support myelination. This includes proteins like Cadm1. However, whether these sites of adhesion are truly "synapse-like" is still unclear. For example, in Figure 6, the authors visualize Scarlet-labeled Synaptophysin in axons, and they do not see any significant difference in the "colocalization coefficient" between Scarlet puncta and control or dnCadm1b-expressing myelin sheaths. It is also possible that these proteins are playing an adhesive role in myelin formation. Many cells use adhesion proteins for cell-cell interactions, so this is not strong

evidence of functional similarity to a synapse.

At the very least, the authors should dial back their language to be clear that the data, while consistent with this "synapse" hypothesis, do not prove it. For example the title, "Synaptic proteins expressed by oligodendrocytes mediate CNS myelination" is not entirely accurate. Synaptic adhesion proteins like Cadm1 are found to mediate CNS myelination. But the other parts of the paper / elements of synapses they discover (vesicular release in axon, PSD-95 in oligodendrocyte) are not shown here to "mediate CNS myelination," just to exist in the context of myelin. And I disagree with the last sentence of the abstract that states, "Our work reveals shared mechanisms of synaptic and myelin plasticity." "Suggests" would be more appropriate here.

Together, I recommend that the authors, in the abstract and discussion of their paper, be very forthright about what their data allow them to conclude. I would also like a paragraph of the discussion to discuss these caveats I outline above. This will not undercut the paper at all but draw attention to all of the interesting future directions to be pursued based on the observations in this paper.

Reviewer #3 (Remarks to the Author):

The reviewers have largely satisfied my concerns.

There is one significant area of concern remaining. The authors state as evidence for the resolution of their by stating that the approach was developed by a company. Manufacture stated resolutions are typically based on ideal conditions and samples. Thus the authors do not know what the resolution of their system is or whether it can achieve the same resolution. Similar concerns are raised for the citation. It is true that this approach can provide the level of resolution claimed, but does the instrument that they are using?

Ideally the authors would simply test their instrument on a sample such as two sub-diffraction sized beads and measure the resolution of there system e.g. what is the minimum apparent distance between the two bead before they appear to be a single object. This would give the maximum resolution in an ideal sample. They should also measure the minimum distance between two objects in zebra fish that can be discriminated. Here using a single color (really they should test each color used), what is the minimum distance were they can tell a spot is two spots? This would be useful information for the field, and might temper the authors conclusions.

Minor:

I am still confused by the term "impartial". Since it means equitable, even-handed, or unbiased. I guess that the authors are trying to imply that the labeling is everywhere. But there must be less loaded words to use that would better describe the organization. How about "even", "sustained", "consistent" or "uniform"? To me these all are better descriptors of the data.

Reviewer #4 (Remarks to the Author):

In my first review of the manuscript "Synaptic proteins expressed by oligodendrocytes mediate CNS myelination", I acknowledged the novelty and potential importance of providing experimental evidence of how active axonal signalling may regulate myelin growth. I had two major criticisms:

- the first was that no experiments have been carried out to test a requirement for the proposed synaptic axo-glia signalling machinery for axon activity driven myelination (see my initial general

assessment, and specific point #1). No such experiments have been performed for the revised version, which I think would have been critical to provide functional evidence for this signalling machinery (because activity dependent signalling is what synapses do). The finding that mutant *cadm1b* affects sheath length does not test this, because any effect could also result from altered adhesion and be independent of any axo-glia synaptic signalling.

I had also commented on the *cadm1b* phenotype and suggested to provide additional data points to strengthen the robustness of this phenotype (see my initial comments #4 and #5). The authors now provide one new image of a *dn-cadm1b* expressing oligodendrocyte at 8dpf, which also shows some short sheaths (but not quantified). The authors argue that intrinsic differences of cell age are unlikely because differences between groups are bigger than within a group, plus their conservative statistical comparison. However, how have individuals for each group been selected in such case? My understanding is that one of the major advantages of zebrafish studies is the ability to analyse single cells over time. Here, a few additional data points showing the same cell, for example at P3 and P5, would have been sufficient to clarify this concern and to strengthen their conclusions.

- the second major set of questions related to the need to provide stronger, more convincing evidence of the validity of the reagents used here. Here, the authors addressed my points #2 and #3, for example by immunocytochemistry for PSD95 (as suggested). However, none of these data have been quantified to exclude that the observations presented did not occur by chance. This could have easily been done (e.g. by quantification of PSD95 co-localisation in myelin sheaths before and after randomization of PSD95 punctae). This would have been important, particularly because the proposed localization of transgenically expressed proteins to sheath ends is still not fully convincing). Images showing membrane targeted GFP in Fig 4c, 6h2 and 6h3, as well as cytosolic *sox10:tagRFP* (Fig5b) also sometimes appear brighter at sheath ends, just as it was proposed for the localization of PSD95, *cadm1b*, and *casca* to sheath ends (see Fig 3b, Fig5c, and Fig5d respectively). These data need an unbiased quantification method.

Thank you for the opportunity to respond to the Reviewer comments and revise our manuscript. Our responses to the comments appear below.

Reviewer #2

Comments (paraphrased). The manuscript does not present “solid evidence that oligodendrocyte axon interaction is functionally similar to a neuronal synapse” because we do not show that (a) oligodendrocytes respond to axon vesicle fusion, (b) PSD95/CASK are functionally important to myelin sheath formation and (c) synaptogenic adhesion molecules mediate axo-glia signaling rather than simply promote adhesion.

Response. These all are extremely valid points that we hope to address with our ongoing and future studies. However, these are not simple problems to solve and we just do not have the tools in place to solve them at this time. The Reviewer very kindly offered us the opportunity to make revisions to the text instead of requiring additional experiments. We have taken that opportunity. Specifically, we changed the title, slightly modified the Abstract and Introduction and included a new paragraph in the Discussion to better describe the caveats of our study. The new paragraph is reproduced here:

The presence of these canonical synaptic structures at the axon-myelin interface is consistent with the possibility that these proteins function similarly as they do at neuronal synapses, participating in exocytosis of neurotransmitter and anchoring of receptors to support communication. However, other functions are also possible for these features. For example, “synaptic” vesicles are identified, by us and by others, by the presence of membrane proteins such as Synaptophysin (Syp) that are generally absent in non-synaptic vesicles. A limitation of this classification is that Syp presence in a vesicle membrane indicates nothing about the cargo contained. It is possible that the Syp-labeled vesicles docked under myelin represent a distinct pool from the recycling pool that is shared among presynaptic boutons⁵⁹, and may contain distinct cargo intended for myelin, such as pro-myelinating factors. A similar argument applies to our observation that canonical postsynaptic scaffolds, including PSD95 and CASK, are present in myelin sheaths. While these scaffolds could be anchoring receptors and signaling molecules as they do at synapses, myelinating oligodendrocytes downregulate expression of many receptors, raising the possibility that these scaffolds anchor other proteins or serve other functions. One functional similarity that we did observe is that of the synaptogenic adhesion molecules: interruption of these proteins in neurons disrupts synapse size and number, and we observed a reduction in myelin sheath length. These proteins are sufficient for presynaptic assembly when expressed in non-neuronal cells. However, we observed a severe reduction in myelin sheath length upon expression of dominant-negative that did not diminish presynaptic vesicle clustering, indicating that adhesion may be a more important function for these molecules at the axon-myelin interface.

Reviewer #3

Comment. There is one significant area of concern remaining. The authors state as evidence for the resolution of their by stating that the approach was developed by a company. Manufacture stated resolutions are typically based on ideal conditions and samples. Thus the authors do not know what the resolution of their system is or whether it can achieve the same resolution. Similar concerns are raised for the citation. It is true that this approach can provide the level of resolution claimed, but does the instrument that they are using?

Ideally the authors would simply test their instrument on a sample such as two sub-diffraction sized beads and measure the resolution of their system e.g. what is the minimum apparent distance between the two bead before they appear to be a single object. This would give the maximum resolution in an ideal sample. They should also measure the minimum distance between two objects in zebra fish that can be discriminated. Here using a single color (really they should test each color used), what is the minimum distance were they can tell a spot is two spots? This would be useful information for the field, and might temper the authors conclusions.

Response. We appreciate Reviewer 3's suggestion to determine the resolution of SRRF-analyzed images acquired on our microscope. The SRRF algorithm was designed to improve resolution achievable by conventional microscopy approaches (Henriques et al, 2016, Nat Commun). As noted by Reviewer 3, the benefits of SRRF could be limited by acquisition. We consulted with our Advanced Light Microscopy Core to

determine the resolution we could acquire with SRRF on our microscope. Following the advice of the Core, we measured the resolution of SRRF images with Fourier ring correlation (FRC), the current standard for determining resolution of super-resolution images (Nieuwenhuizen et al, 2013, Nature Methods; Banterle et al, 2013, J Struct Biol). We measured resolution of SRRF images of both sub-diffraction (100 nm) beads suspended in agarose, similar to zebrafish live imaging, and also of SRRF images of cells labeled in zebrafish, to estimate the resolution of images that appear in the manuscript (Fig. 5c, d) (see attached Figure). For these analyses, we acquired two sets of single optical section stacks of 75-110 frames. We then processed each set with SRRF to generate super-resolution reconstructions of each set. Then, we subjected the two SRRF reconstructions to FRC, which compares the two SRRF reconstructions to each other to determine the resolution.

We were advised by the Core that super-resolution imaging of beads would likely overestimate the resolution that SRRF achieves in biological samples. At Reviewer 3's recommendation we tried it anyway. By 3-sigma FRC, our acquisition protocol paired with SRRF achieved ~71 nm resolution when imaging agarose-suspended beads. This likely overshoots the true resolution because beads are featureless and easily over-reduced to single points. The FRC-determined resolution was tempered in live cells in zebrafish. According to 3-sigma Fourier ring correlation, our acquisition protocol paired with SRRF processing achieved ~123 nm resolution in cells labeled in the zebrafish. This represents a significant improvement over standard confocal microscopy (280 nm on our 63x water objective) and is comparable to what SRRF was previously documented to produce at this frame rate (100-140 nm) (Henriques et al, 2016, Nat Commun) and is similar to other super-resolution imaging approaches, such as structured illumination microscopy (~125 nm).

BEADS IN AGAROSE (100 nm)

CELLS IN ZEBRAFISH *Tg(sox10:mRFP)*

Fourier ring correlation (FRC) analysis of resolution achieved by super-resolution radial fluctuation (SRRF) processing. Top, single optical sections (75 frames) of fluorescent 100 nm beads suspended in agarose were acquired in two independent acquisitions to generate SRRF reconstructions. The SRRF reconstructions were compared by 3-sigma FRC analysis to generate FRC curves and resolution values in both pixels and nanometers. Bottom, single optical

sections (109 frames) of oligodendrocytes in a live *Tg(sox10:mRFP)* agarose-mounted larva acquired twice and processed with SRRF. The SRRF reconstructions were compared by 3-sigma FRC analysis to generate FRC curves and resolution values in both pixels and nanometers.

Comment. I am still confused by the term "impartial". Since it means equitable, even-handed, or unbiased. I guess that the authors are trying to imply that the labeling is everywhere. But there must be less loaded words to use that would better describe the organization. How about "even", "sustained", "consistent" or "uniform"? To me these all are better descriptors of the data.

Response. We changed "impartial" to "uniform" because it is indeed a better description of the labeling pattern.

Reviewer #4

Comment. - the first was that no experiments have been carried out to test a requirement for the proposed synaptic axo-glial signalling machinery for axon activity driven myelination (see my initial general assessment, and specific point #1). No such experiments have been performed for the revised version, which I think would have been critical to provide functional evidence for this signalling machinery (because activity dependent signalling is what synapses do). The finding that mutant *cadm1b* affects sheath length does not test this, because any effect could also result from altered adhesion and be independent of any axo-glial synaptic signalling.

Response. This is similar to points raised by Reviewer #2, which we addressed above.

Comment. I had also commented on the *cadm1b* phenotype and suggested to provide additional data points to strengthen the robustness of this phenotype (see my initial comments #4 and #5). The authors now provide one new image of a *dn-cadm1b* expressing oligodendrocyte at 8dpf, which also shows some short sheaths (but not quantified). The authors argue that intrinsic differences of cell age are unlikely because differences between groups are bigger than within a group, plus their conservative statistical comparison. However, how have individuals for each group been selected in such case? My understanding is that one of the major advantages of zebrafish studies is the ability to analyse single cells over time. Here, a few additional data points showing the same cell, for example at P3 and P5, would have been sufficient to clarify this concern and to strengthen their conclusions.

Response. We provide 8 dpf data as new Figure panels 6k and l.

Comment. - the second major set of questions related to the need to provide stronger, more convincing evidence of the validity of the reagents used here. Here, the authors addressed my points #2 and #3, for example by immunocytochemistry for PSD95 (as suggested). However, none of these data have been quantified to exclude that the observations presented did not occur by chance. This could have easily been done (e.g. by quantification of PSD95 co-localisation in myelin sheaths before and after randomization of PSD95 punctae). This would have been important, particularly because the proposed localization of transgenically expressed proteins to sheath ends is still not fully convincing). Images showing membrane targeted GFP in Fig 4c, 6h2 and 6h3, as well as cytosolic *sox10:tagRFP* (Fig5b) also sometimes appear brighter at sheath ends, just as it was proposed for the localization of PSD95, *cadm1b*, and *casca* to sheath ends (see Fig 3b, Fig5c, and Fig5d respectively). These data need an unbiased quantification method.

Response. We performed a randomization analysis and show the data below. This analysis indicates that co-localization of PSD95, detected by immunofluorescence, with myelin sheaths did not occur by chance.

Costes randomization analysis of PSD95 immunofluorescence colocalization with *Tg(mbpa:eGFP-CAAX)* fluorescence. Costes colocalization analysis (Costes et al, 2004, Biophysical Journal) of fluorescent signal between anti-PSD95 labeled with AF647 and transgenic reporter *Tg(mbpa:eGFP-CAAX)* before (original) or after (randomized) performing 1000 rounds of randomization per image for n=20 images containing both channels. Analysis was performed with the Fiji plugin JACoP with specific parameters: confocal image, bin width=0.001, pixel size=0.174 μm , randomizing in both the xy and z directions. Data are presented pairwise, analyzed by paired Wilcoxon, and the value plotted for the randomized condition is the maximum value from the output range (e.g., 0 +/- 0.003 is plotted as +0.003).

REVIEWERS' COMMENTS:

Reviewer #2 (Remarks to the Author):

The authors have satisfactorily responded to my previous comments and suggestions.

Reviewer #3 (Remarks to the Author):

The authors have responded effectively to my comments providing the necessary measurements and characterization of their instrument. In addition, it appears that they have responded effectively to the other reviewer's comments. This manuscript should add significantly to the literature and will be of interest to many.

Reviewer #4 (Remarks to the Author):

- I had criticized that the authors do not provide evidence that functional synaptic signaling occurs and is required for sheath extension (similar to reviewer #2). The modified manuscript title and text now take this circumstance appropriately into account.
- The quantification of co-localization between pre-synaptic neuronal punctae and myelin sheaths (original versus randomized) are important and convincing data, and should thus be added to the manuscript as supplementary information.

To the editors,

Please find here our detailed responses to reviewer comments.

Reviewer #4 (Remarks to the Author):

- The quantification of co-localization between pre-synaptic neuronal punctae and myelin sheaths (original versus randomized) are important and convincing data, and should thus be added to the manuscript as supplementary information.

We now provide this as Supplementary Figure 1. Because we now include Supplementary Information, we also now included data documenting the resolution of the SRRF imaging technique, originally requested by Reviewer 2. We also moved the oligonucleotide list to Supplementary Material.